# Addressing erroneous scale assumptions in microbe and gene set enrichment analysis

Kyle C. McGovern[1], Michelle Pistner Nixon[2], Justin D. Silverman [1,2,3,4]*

**1** Program in Bioinformatics and Genomics, Pennsylvania State University, State College, Pennsylvania, United States of America, **2** College of Information Sciences and Technology, Pennsylvania State University, State College, Pennsylvania, United States of America, **3** Departments of Medicine and Statistics, Pennsylvania State University, State College, Pennsylvania, United States of America, **4** Institute for Computational and Data Science, Pennsylvania State University, State College, Pennsylvania, United States of America

\* JustinSilverman@psu.edu

**Data Availability Statement:** All data used in this manuscript has been previously published and is publicly available. The thyroid and breast tissue datasets are available under accession numbers GSE86354 and GSE62944 in NCBI's Gene

## Abstract

By applying Differential Set Analysis (DSA) to sequence count data, researchers can determine whether groups of microbes or genes are differentially enriched. Yet sequence count data suffer from a *scale limitation*: these data lack information about the scale (i.e., size) of the biological system under study, leading some authors to call these data compositional (i.e., proportional). In this article, we show that commonly used DSA methods that rely on normalization make strong, implicit assumptions about the unmeasured system scale. We show that even small errors in these *scale assumptions* can lead to positive predictive values as low as 9%. To address this problem, we take three novel approaches. First, we introduce a sensitivity analysis framework to identify when modeling results are robust to such errors and when they are suspect. Unlike standard benchmarking studies, this framework does not require ground-truth knowledge and can therefore be applied to both simulated and real data. Second, we introduce a statistical test that provably controls Type-I error at a nominal rate despite errors in scale assumptions. Finally, we discuss how the impact of scale limitations depends on a researcher's scientific goals and provide tools that researchers can use to evaluate whether their goals are at risk from erroneous scale assumptions. Overall, the goal of this article is to catalyze future research into the impact of scale limitations in analyses of sequence count data; to illustrate that scale limitations can lead to inferential errors in practice; yet to also show that rigorous and reproducible scale reliant inference is possible if done carefully.

## Author summary

A common task in the analysis of DNA sequence count data is to determine whether sets of biologically related genes or microbes are differentially enriched between two experimental conditions (Differential Set Analysis; DSA). Yet DSA can be confounded by the non-biological (i.e., technical) variation in sequencing depth. To address this issue, many researchers use normalization techniques to remove this variation. The choice of

Expression Omnibus (GEO). The NYC Health and Nutrition Examination Survey smoker oral microbiome dataset is available through the 'nychanesmicrobiome' R package. All code needed to reproduce the analyses in this work is provided at: https://github.com/kyle-mcgovern/DSAScaleError.

**Funding:** J.D.S and M.P.N were supported in part by the National Institute of General Medical Sciences (NIH 1R01GM148972-01). K.C.M was supported in part by the Computational, Biology, and Statistics (CBIOS) fellowship training program (NIH 5T32GM102057-10). The funders had no role in study design, data collection and analysis, decision to publish, or preparation of the manuscript.

**Competing interests:** The authors have declared that no competing interests exist.

normalization can dominate modeling results yet we lack tools for properly validating this decision. Here we develop statistical and computational tools that allow researchers to quantify the robustness of analytical results to the choice of normalization. These methods aim to improve the rigor and reproducibility of commonly performed set enrichment analyses.

## Introduction

Sequence count data (e.g., data from 16S rRNA-Seq or RNA-Seq studies) have become ubiquitous in modern biomedical research. These data are often used to identify whether a pre-determined set of entities (i.e., set of microbes or set of genes) are differentially enriched between two biological conditions [1, 2]. For example, in the analysis of RNA-Seq data, researchers often use tools such as Gene Set Enrichment Analysis (GSEA) [1] to identify pathways (sets of genes) that are up or down regulated between conditions [3–5]. We refer to this inferential task as Differential Set Analysis (DSA). We show that common approaches to DSA can display high rates of false positives due to limitations of the sequence counting measurement process. We introduce new tools and insights to help researchers mitigate these errors.

Sequencing depth (the number of measured DNA molecules in a sample) can vary substantially between samples due to non-biological (i.e., technical) factors [6, 7]. This variation can confound analyses. To remove this unwanted variation and facilitate inference that requires between sample comparisons, many authors turn to normalization [8, 9]. Yet there are limitations to this approach. Various methods of normalization have been proposed, but current guidelines for choosing a normalization primarily rely on simulation-based benchmarking studies which may not generalize to a particular analysis of real data (e.g., [10]). This problem is magnified by the fact that the choice of normalization can drive modeling results [11, 12]. Yet other perspectives on the non-biological variation of sequencing depth bring the use of normalization into question. Rather than viewing this non-biological variation as *extra noise* that must be removed, some view it as a symptom of an imperfect measurement process that *lacks information* about the scale (i.e., total size) of the system being studied [7, 13, 14]. From this perspective, any normalization is actually making an implicit assumption about the system scale (a *scale assumption*) that we cannot validate from the observed data [15]. Following this view, some authors argue against the use of any normalization, instead arguing that we should only use these data to answer questions that are invariant to the unmeasured system scale [7, 13]. This view is predominant within the field of Compositional Data Analysis (CoDA) which is founded upon the axiom of scale invariance [16]. While intuitive, this scale invariance perspective can be scientifically limiting, as many biologically relevant scientific questions require scale information [17, 18].

A third perspective has recently emerged. Nixon et al. proposed a reformulation of CoDA called *Scale Reliant Inference* (SRI) for statistically rigorous non-scale invariant (scale reliant) analyses with sequence count data [14]. Using SRI, those authors show that statistically rigorous analysis of these data requires explicitly considering uncertainty in the unmeasured system scale: in essence building statistical models that consider uncertainty in the normalization method itself. By applying their framework to the study of Differential Analysis (DA, i.e., differential abundance and differential expression analyses) they show that even slight errors in scale assumptions can lead to false discovery rates as high as 80%. Viewing DSA as a generalization of DA, where the goal is to study sets of entities rather than individual entities, we hypothesized that errors in scale assumptions may lead to inferential errors in DSA.

We review concepts and terminology from the field of SRI to study how errors in scale assumptions can impact DSA. As in SRI, our approach centers around a mathematical quantity called a *target estimand*, the quantity a researcher wants to estimate. We study the impact of erroneous scale assumptions and characterize how this impact depends on the target estimand. We show that many common estimands are scale reliant and that, in the context of those estimands, common methods for performing DSA result in elevated rates of false positives. To mitigate these problems, we develop a type of sensitivity analysis to identify possible false positives. In addition, we characterize a broad class of target estimands that are insensitive to erroneous scale assumptions and discuss how to determine when research goals fall within this class. Beyond providing new tools for performing DSA, the overall purpose of this article is to catalyze future research and discussions into the impact of scale limitations in sequence count data; to show that rigorous and reproducible scale reliant inference is possible with these data; and to also highlight that care is required to perform such analyses.

## Review of scale reliant inference

Scale Reliant Inference (SRI) is a conceptual framework used to study scale assumptions in sequence count data [14]. This section reviews core terms and concepts from SRI relevant to the study of DSA. For concreteness, this review is presented in terms of a hypothetical study performing differential abundance analysis on a cross-sectional microbiome dataset of $N$ study participants, half with a disease of interest and half healthy controls. A summary of mathematical notations used throughout this work is provided in Table 1.

**The scaled system, observed data, and target estimand.** For differential abundance, the observed data are represented as a $D \times N$ matrix of counts $Y$ with elements $Y_{dn}$ representing the number of DNA sequences that map to the $d$-th microbial taxon in the sample obtained from participant $n$. Central to SRI is the notion that the observed data ($Y$) is a measurement of an underlying biological system called a *scaled system* and denoted as $W$. Like $Y$, $W$ is a $D \times N$ matrix. Unlike $Y$, the elements $W_{dn}$ represent the true (as opposed to measured) amount of the $d$-th taxon in the gut microbiota of the $n$-th participant. Note that the definition of *true amount* is not restricted to the total number of microbes in a subject's gut. For example, for a

**Table 1. Summary of mathematical notation.**

| Symbol | Definition |
|---|---|
| $d$ | Entity (e.g., microbe or gene) index; $d = 1, \ldots, D$. |
| $n$ | Sample index; $n = 1, \ldots, N$. |
| $W_{dn}$ | An element of the $D \times N$ matrix $W$ representing the true amount of entity $d$ in the system from which sample $n$ was obtained. |
| $Y_{dn}$ | An element of the $D \times N$ matrix $Y$ representing the number of DNA sequences observed from entity $d$ in sample $n$. |
| $\perp$ | The scale component of a quantity, e.g., $W_n^{\perp} = \sum_{d=1}^{D} W_{dn}$. |
| $\parallel$ | The compositional (i.e., proportional) component of a quantity, e.g., $W_{dn}^{\parallel} = W_{dn} / W_n^{\perp}$. |
| $\theta_d$ | An estimand; the Log Fold Change (LFC) in abundance of entity $d$. |
| $\vartheta_d$ | An estimand; the Log Fold Change in Centered Log-Ratio (CLR) normalized abundance of entity $d$. |
| $\phi_S$ | An estimand; the enrichment of entity set $S$; $\phi_S = \pm 1$ if $S$ is differentially enriched/depleted, and $\phi_S = 0$ if $S$ is not differentially enriched/depleted. |
| $\psi$ | An estimand; we use $\psi$ to discuss estimands abstractly without focusing on a particular estimand. |
| $\hat{\phantom{x}}$ | An estimate of a quantity, e.g., $\hat{\theta}_d$ is an estimate of $\theta_d$. |
| $\epsilon_d$ | Error in an LFC estimate; i.e., $\theta_d = \hat{\theta}_d + \epsilon_d$. |

researcher interested in host-pathogen interactions, true amounts could be the ratio of bacterial to human cells at the epithelial surface of the distal colon.

The scaled system represents an unmeasured standard of truth, a reference against which the limitations of the observed data can be discussed. While the sequence counting process can provide limited information about the scaled system in many ways (e.g., PCR bias [19]), this article studies the lack of scale information in these data. Consider that the scaled system can be uniquely described in terms of its scale (summed, $W^\perp$) and compositional (proportional, $W^\parallel$) parts:

$$W_{dn} = W_{dn}^\parallel W_n^\perp \tag{1}$$

$$W_n^\perp = \sum_{d=1}^{D} W_{dn}$$

$$W_{dn}^\parallel = \frac{W_{dn}}{W_n^\perp}.$$

Intuitively, saying sequence count data lacks scale information means that, due to the non-biological variation in sequencing depth, sequence count data cannot be used to estimate the system scale ($W^\perp$) to any reasonable degree of precision [6, 18] (see Nixon et al. for a more formal definition involving model identifiability [14]).

Ultimately, the extent to which scale limitations matter to an applied researcher depends on their research goal. In SRI, the research goal is represented by a mathematical quantity called a target estimand. Formally, a target estimand ($\psi$) is some function $f$ applied to the scaled system: $\psi = f(W)$. For example, a differential abundance analysis may estimate the Log-Fold-Change (LFC) of each taxa $d$:

$$\theta_d = \operatorname*{mean}_{n:x_n=1} (\log W_{dn}) - \operatorname*{mean}_{n:x_n=0} (\log W_{dn}). \tag{2}$$

for a binary covariate $x_n$ denoting two mutually exclusive conditions (e.g., case versus control).

The fundamental challenge of SRI occurs when the desired target estimand relies upon scale information not present in the observed data. For example, using the relationship $\log W_{dn} = \log W_{dn}^\parallel + \log W_n^\perp$ implied by Eq (1), the LFC target estimand in Eq (2) can be written as

$$\theta_d = \underbrace{\left[\operatorname*{mean}_{n:x_n=1}(\log W_{dn}^\parallel) - \operatorname*{mean}_{n:x_n=0}(\log W_{dn}^\parallel)\right]}_{\theta_d^\parallel} + \underbrace{\left[\operatorname*{mean}_{n:x_n=1}(\log W_n^\perp) - \operatorname*{mean}_{n:x_n=0}(\log W_n^\perp)\right]}_{\theta^\perp} \tag{3}$$

$$= \theta_d^\parallel + \theta^\perp.$$

In vector notation, this can be written as $\theta = \theta^\parallel + \mathbf{1}\theta^\perp$ where $\theta = (\theta_1, \ldots, \theta_D)$, $\theta^\parallel = (\theta_1^\parallel, \ldots, \theta_D^\parallel)$, and where $\mathbf{1}$ denotes a vector of ones. This target estimand is scale reliant: it requires knowledge of $\theta^\perp$ and therefore $W^\perp$ to uniquely determine $\theta_d$.

**Scale assumptions and the applied estimator.**   The target estimand, the scaled system, and the observed data make up the three core constructs of SRI. From this core, the goal of an analysis can be defined, and the challenge of achieving that goal given the limited information content of the observed data can be investigated. Notably, this core does not include the specific analytical method applied. That method, referred to as an applied estimator $g$, is a function applied to the observed data to estimate the target estimand $\psi$: $\hat{\psi} = g(Y)$. In SRI, the

target estimand serves as a backdrop against which the impact of errors made by the applied estimator can be studied. For example, consider observed data ($Y$) that lacks scale information and a target estimand ($\psi$) that is scale reliant (that requires scale information). Against this backdrop, it is clear that *any* applied estimator $\hat{\psi} = g(Y)$ that produces a unique estimate of $\psi$ must have made some assumption about the unmeasured scale of the system $W^{\perp}$. In SRI, these are called *scale assumptions*.

Following Nixon et al. [14], we illustrate the concept of scale assumptions through a study of the Centered Log-Ratio (CLR) normalization in differential abundance analysis. Recognizing the scale limitations of sequence count data, a variety of tools (e.g., ALDEx2 [20] or Songbird [15]) estimate LFCs using CLR normalized abundances. In brief, these methods can be thought of as producing estimates of a target estimand:

$$\vartheta_d = \operatorname*{mean}_{n:x_n=1}(\log W_{dn}^{clr}) - \operatorname*{mean}_{n:x_n=0}(\log W_{dn}^{clr}) \tag{4}$$

where $W_{dn}^{clr} = \log\left[W_{dn}/G(W_{\cdot n})\right]$ and where $G(W_{\cdot n})$ denotes the geometric mean of the vector $W_{\cdot n} = (W_{1n}, \ldots, W_{Dn})$. The system's scale (i.e., $W^{\perp}$) cancels out of the fraction $\log[W_{dn}/G(W_{\cdot n})]$ (see Section F in S1 Text), and thus $W_{dn}^{clr}$ can be equivalently expressed entirely in terms of the system's composition (i.e., $W^{\parallel}$): $W_{dn}^{clr} = \log\left[W_{dn}^{\parallel}/G(W_{\cdot n}^{\parallel})\right]$. The CLR target estimand may be further decomposed into its compositional and scale components, similar to Eq (3), as

$$
\begin{aligned}
\vartheta_d &= \underbrace{\left[\operatorname*{mean}_{n:x_n=1}(\log W_{dn}^{\parallel}) - \operatorname*{mean}_{n:x_n=0}(\log W_{dn}^{\parallel})\right]}_{\vartheta_d^{\parallel}} + \\
&\qquad \underbrace{\left[\operatorname*{mean}_{n:x_n=1}\left(\log \frac{1}{G(W_{\cdot n}^{\parallel})}\right) - \operatorname*{mean}_{n:x_n=0}\left(\log \frac{1}{G(W_{\cdot n}^{\parallel})}\right)\right]}_{\vartheta^{\perp}} \\
&= \vartheta_d^{\parallel} + \vartheta^{\perp}.
\end{aligned}
\tag{5}
$$

The CLR target estimand in Eq (4) is not the same as the LFC target estimand in Eq (2). Yet many authors take the estimates produced by ALDEx2 and Songbird as estimates of the LFC as defined in Eq (2) (e.g., [21, 22]). While $\theta^{\parallel} = \vartheta^{\parallel}$, $\theta^{\perp}$ (Eq 3) differs from $\vartheta^{\perp}$ (Eq 5) as the former is a function of the system's scale (i.e., $W^{\perp}$) and the latter is a function of the system's composition (i.e., $W^{\parallel}$). Assuming that the output of ALDEx2 or Songbird represents estimates of LFCs (as defined in Eq 2) is equivalent to an implicit assumption that $\theta_d = \vartheta_d$. This assumption can be further simplified to $\theta^{\perp} = \vartheta^{\perp}$ which is an assumption that the true log fold change in scale can be imputed from the system composition:

$$\theta^{\perp} = \operatorname*{mean}_{n:x_n=1}\left(\log \frac{1}{G(W_{\cdot n}^{\parallel})}\right) - \operatorname*{mean}_{n:x_n=0}\left(\log \frac{1}{G(W_{\cdot n}^{\parallel})}\right). \tag{6}$$

We refer to this as the CLR assumption.

To date, SRI has primarily been used to study target estimands associated with differential analysis. In what follows, we provide the first application of SRI to DSA.

## Results

### Conceptual overview of methods

For a set of entities $S$, DSA estimates a target estimand $\phi_S$ which can take on three values: +1 if the set is differentially enriched, −1 if differentially depleted, and 0 if neither enriched nor depleted. DSA is often performed in two steps: first, LFCs are estimated using tools like ALDEx2 [20] or DESeq2 [23]; second, tools like Gene Set Enrichment Analysis (GSEA) [1] are applied to the estimated LFCs to produce an estimate $\hat{\phi}_S$. In the next section, we introduce these two-step estimators using the language of applied estimators and target estimands. After that, we provide a core methodological contribution of this work: we show how errors in scale assumptions used to estimate LFCs propagate into errors in estimates of $\phi_S$. This result forms the basis of our LFC Sensitivity Analyses which allow researchers to quantify the sensitivity of estimates $\hat{\phi}_S$ to errors in scale assumptions. This idea also forms the basis of the LFC Sensitivity Test which identifies entity sets where conclusions about enrichment or depletion are completely insensitive (invariant) to errors in scale assumptions. Beyond the main text, Section A in S1 Text develops more general forms of sensitivity analysis for DSA that generalize beyond these two-step LFC-based estimators and can be applied to other methods like CAMERA [24].

### Applied estimators and target estimands for DSA

There are many applied estimators for DSA. A particularly popular approach is to apply Gene Set Enrichment Analysis (GSEA) [1] to LFCs estimated using tools such as ALDEx2 [20], Songbird [15], or DESeq2 [23]. DSA estimators such as GSEA and others [25] can be thought of as two-stage applied estimators: the first stage estimator $h$ (e.g., ALDEx2) estimates LFCs from observed data ($\hat{\theta} = h(Y)$) and the second stage estimator $u$ (e.g., GSEA) then estimates DSA using $\hat{\theta}$ ($\hat{\phi}_S = u(\hat{\theta})$). We use the two-stage form of these applied estimators to define target estimands for DSA.

It is challenging to identify target estimands. Just as researchers' goals can differ, so too can their definitions of enriched and depleted entity sets. Moreover, many studies do not explicitly state their estimation goals but instead simply use an applied estimator leaving the estimation goal implicit. To address this challenge, we assume a correspondence between the applied estimator a researcher uses and their estimation goals. Consider a researcher who uses a DSA applied estimator with second stage $\hat{\phi}_S = u(\hat{\theta})$. We assume their target estimand is defined just as the applied estimator but with the estimated LFCs replaced with the true LFCs: $\phi_S = u(\theta)$.

### LFC sensitivity analysis and testing

Consider any DSA applied estimator that can be written as $\hat{\phi}_S = u(\hat{\theta})$ with corresponding target estimand $\phi_S = u(\theta)$. We can relate the true value of the estimand $\phi_S$ and the estimate $\hat{\phi}_S$ by considering error in the estimated value of $\theta$. Let $\epsilon$ denote error in the estimate $\hat{\theta}$ such that $\theta = \hat{\theta} + \epsilon$. Just as the LFC estimate can be decomposed as $\theta = \theta^{\parallel} + \mathbf{1}\theta^{\perp}$, the error can also be decomposed as $\epsilon = \epsilon^{\parallel} + \mathbf{1}\epsilon^{\perp}$ where $\epsilon^{\parallel} = \theta^{\parallel} - \hat{\theta}^{\parallel}$ is a $D$ vector with elements $\epsilon_d^{\parallel}$ denoting error in each compositional component of the estimate, and $\epsilon^{\perp} = \theta^{\perp} - \hat{\theta}^{\perp}$ is a scalar denoting error in the estimated scale. While there can be error in compositional estimation such that, for some $d$, $\epsilon_d^{\parallel} \neq 0$, the consideration of such error only serves to complicate our study of scale. Consider that unless $\epsilon^{\perp}$ and $\epsilon_d^{\parallel}$ are strongly anti-correlated, if a conclusion is highly sensitive to error $\epsilon^{\perp}$ when $\epsilon_d^{\parallel} = 0$, it will still be highly sensitive when $\epsilon_d^{\parallel} \neq 0$. Thus, we restrict our analysis

to $\epsilon^{\perp}$ by assuming that $\epsilon_d^{\parallel} \ll \epsilon^{\perp}$ such that $\epsilon \approx \mathbf{1}\epsilon^{\perp}$. Then the true value $\theta$ can be expressed in terms of the estimate $\hat{\theta}$ as

$$\theta \approx \hat{\theta} + \mathbf{1}\epsilon^{\perp}. \tag{7}$$

Returning to DSA, Eq (7) can be used to represent the truth ($\phi_S$) in terms of error $\epsilon^{\perp}$ and LFC estimates $\hat{\theta}$:

$$\begin{aligned} \phi_S \quad &= u(\theta) \\ &\approx u(\hat{\theta} + \mathbf{1}\epsilon^{\perp}). \end{aligned} \tag{8}$$

By comparing $\phi_S$ versus $\hat{\phi}_S$ as a function of $\epsilon^{\perp}$ we can study how our estimates differ from the truth as a function of errors in scale assumptions ($\epsilon^{\perp}$). We call this LFC Sensitivity Analysis.

LFC Sensitivity Analysis has a number of appealing properties. To perform LFC Sensitivity Analysis, we only need to know the applied LFC estimates $\hat{\theta}$ and not the true value of the DSA target estimand $\phi_S$ or the true value of the LFCs $\theta$. Thus this method can be applied to both simulated and real data.

Another appealing quality of LFC Sensitivity Analysis is its interpretability. Consider that $\hat{\theta}^{\perp} = \text{mean}_{n:x_n=1}(\log \hat{W}_n^{\perp}) - \text{mean}_{n:x_n=0}(\log \hat{W}_n^{\perp})$ (Eq 3). This can be rewritten as

$$\hat{\theta}^{\perp} = \log \left[ \frac{G_{n:x_n=1}(\hat{W}_n^{\perp})}{G_{n:x_n=0}(\hat{W}_n^{\perp})} \right]$$

where $G$ denotes the geometric mean. It follows that $\epsilon^{\perp}$ can be interpreted as the error in the assumed log-fold-change of scales. For example, an error of $\epsilon^{\perp} = 1$ can be read as a statement: *the ratio of the mean scale in the case condition compared to the control condition is $e^1 \approx 2.7$ times higher than assumed*. Moreover, this interpretation does not depend on the chosen notion of amount: any units attached to a researcher's chosen notion of amount (e.g., cells per mL) cancel in the ratio.

LFC Sensitivity Analysis can also be used to create a hypothesis test for DSA that is robust to errors in scale assumptions. Suppose there is an entity set $S$ such that for all $\epsilon^{\perp} \in (-\infty, \infty)$ the target estimand $\phi_S$ is always either 1 or −1. Intuitively, conclusions about this entity set are invariant to errors in scale assumptions. We can turn this intuition into a robust hypothesis test for DSA as follows. Let $p_{\epsilon^{\perp}}$ denote a $p$-value corresponding to a test of the null hypothesis $\phi_S = 0$ calculated from a chosen applied DSA estimator when applied to LFCs $\theta = \hat{\theta} + \mathbf{1}\epsilon^{\perp}$. We can calculate a new $p$-value summarizing over $\epsilon^{\perp}$ as

$$p = \max_{\epsilon^{\perp} \in (-\infty, \infty)} p_{\epsilon^{\perp}}. \tag{9}$$

This new $p$-value implicitly defines a test which we call the *LFC Sensitivity Test*. Prior work on Type-I error control in the presence of nuisance parameters (e.g., [26]) establishes that the LFC Sensitivity Test can control Type-I error in spite of errors $\epsilon^{\perp}$ in scale assumptions. Of course the cost of such rigorous Type-I error control is the potential for low statistical power (low probability of detecting non-zero $\phi_S$). Remarkably, in the section *LFC Sensitivity Analysis and Testing of Real Data* we show that the LFC Sensitivity Test displays non-zero power in practice.

### The GSEA-LFC target estimand is typically scale reliant

We focus our study of DSA on GSEA applied to estimated LFCs due to the popularity of this approach. The GSEA algorithm is visualized in Fig 1 and a formal definition is provided in Section B in S1 Text. In brief, GSEA is an algorithm that first orders entities by a ranking statistic (here LFCs; Fig 1A), then measures the degree of non-random clustering in the ranked list through an enrichment score calculated as the maximum distance of a weighted running sum (Fig 1B), and then performs a permutation test to determine whether that enrichment score is unusually large or small. The output of GSEA can be summarized as the quantity $\phi_S$ introduced above.

There are two common variations on GSEA that differ in the permutation scheme used to test the significance of the estimated $\hat{\phi}_S$ statistic. The first permutes the entity labels. The second permutes the sample labels. Assuming adequate sample size, sample-label permutations are generally preferred, as they ensure that the null model accounts for inter-entity correlations [24, 27, 28]. Coupled with these two variants on GSEA, we can define two target estimands each based on applying GSEA to the true LFCs. We denote the estimand formed from GSEA

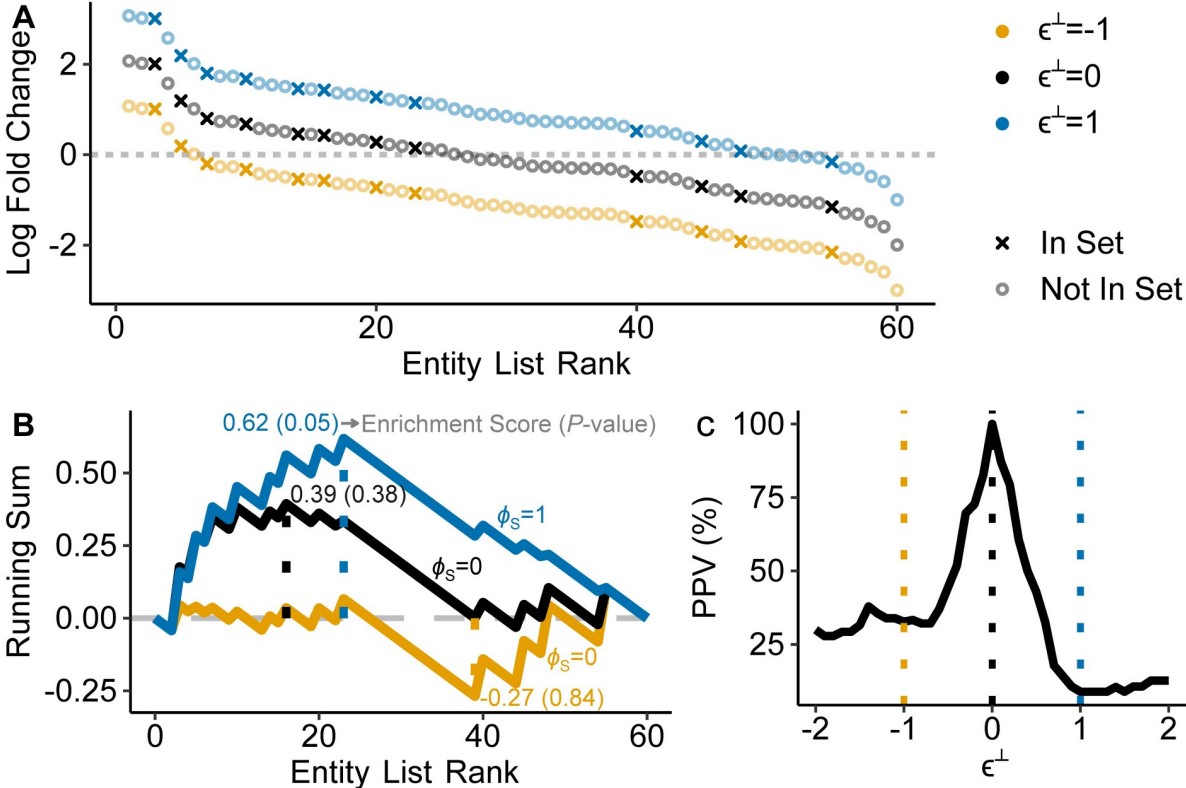

**Fig 1. The GSEA-LFC target estimand is often scale reliant.** (A) The LFCs of 60 entities were simulated from a standard normal distribution and rank ordered (black); 12 of these entities were randomly selected to be in the entity set. For this simulation, the CLR assumption equated to a value of $\hat{\theta}^{\perp} = -0.1$. Blue and orange points represent the true LFCs if the assumed value of $\hat{\theta}^{\perp}$ had error $\epsilon^{\perp}$: $\theta^{\perp} = \hat{\theta}^{\perp} + \epsilon^{\perp}$. (B) For each level of error $\epsilon^{\perp}$ depicted in Panel A, we show the GSEA running sum and the corresponding Enrichment Scores (ES). The dotted lines represents the GSEA enrichment score, which is the maximum distance from zero of the running sum. Also shown are $p$-values and $\phi_S$ values, although they depend on permutation tests which are not depicted visually. $\phi_S$ is the GSEA-LFC estimand. $\phi_S$ is ±1 when the entity set is significantly enriched or depleted (blue line, here $p \leq 0.05$ is used for significance) and 0 when the entity set is not enriched (black and orange lines). (C) 10,000 entity sets of size 12 were simulated using the procedure as in Panel A in order to demonstrates how the Positive Predictive Value (PPV) of GSEA-LFC can vary with $\epsilon^{\perp}$. The dashed lines indicate the same values of $\epsilon^{\perp}$ shown in Panels A and B.

with entity label permutations as GSEA-LFC and the estimand formed from GSEA with sample label permutations as GSEA-LFC-S. In the main text we focus on GSEA-LFC as sensitivity analyses with GSEA-LFC-S are more complex yet show similar results. Still, we present a form of sensitivity analysis for GSEA-LFC-S as well as other DSA methods such as CAMERA [24] in Section A in S1 Text.

We used LFC Sensitivity Analysis to demonstrate that, for many entity sets, the GSEA-LFC target estimand is scale reliant. An example is depicted visually in Fig 1 which shows that different errors $\epsilon^\perp$ in the assumed scale $\hat{\theta}^\perp$ lead to different values of the target estimand $\phi_S$. That is, for the depicted entity set, knowledge of the system composition alone is insufficient to uniquely determine the value of $\phi_S$.

To confirm that these results were not unique to the simulated entity set shown in Fig 1A and 1B, we repeated this analysis with 10,000 simulated entities sets and summarized the results in terms of the Positive Predictive Value (PPV) of the applied GSEA-LFC estimator under various levels of error $\epsilon^\perp$. By design, the PPV equals 100% when the assumed value $\hat{\theta}^\perp$ is equal to the true value $\theta^\perp$ (when $\epsilon^\perp = 0$) but may decrease when $|\epsilon^\perp| > 0$. Fig 1C summarizes those results and shows that the PPV can decrease to below 50% with errors on the order of ±0.5. In words, when the average difference in scales between the two conditions is 1.65 times larger ($e^{0.5}$) or 0.6 times lower ($e^{-0.5}$) than assumed, more than half of the entity sets identified as significantly enriched or depleted are false positives. At $\epsilon = 1$ the PPV drops to just 9%.

Finally, in Section C in S1 Text, we expand upon these simulation studies and show how the PPV of a GSEA-LFC applied estimator varies with different LFC distributions, entity set sizes, and total number of entities. Of these factors, asymmetric LFC distributions (having more entities increase than decrease or vice versa) led to the most striking drops in PPV (to just 0.2%) with only slight error in the assumed scale ($\epsilon^\perp = $ ±0.6). This result reinforces recent work showing the dramatic impact of compositional asymmetry on the fidelity of differential analysis [29].

## LFC sensitivity analysis and testing of real data

We analyzed two previously published studies that used GSEA-LFC applied estimators to perform DSA. The first compared gene pathway expression in healthy versus normal-adjacent-to-tumor thyroid tissue [4]. The second compared the abundance of different microbe sets in the oral microbiota of smokers versus non-smokers [30]. We leave discussion of the microbiome study to Section D in S1 Text, as it resembles the thyroid tissue analysis and simply demonstrates that our conclusions hold beyond gene expression analysis. For both analyses, prior literature was used to determine upper and lower bounds for biologically plausible levels of error ($\epsilon^\perp$) in scale assumptions (see Methods).

The sensitivity of GSEA-LFC to errors in scale assumptions varied substantially over the 50 hallmark gene sets analyzed in Aran et al. [4] (Fig 2). The KRAS Signaling Down pathway was largely insensitive to errors in scale assumptions (it was significant over all $\epsilon^\perp$ tested). Other sets such as the MYC Targets V2 gene set were highly sensitive (significant only over a narrow range of $\epsilon^\perp$). With an error as small as $\epsilon^\perp = -0.05$ multiple gene sets identified as enriched would no longer be significant (e.g., the columns Inflammatory Response to DNA Repair). To interpret the magnitude of this error, consider that for this dataset the CLR assumption equates to $\hat{\theta}^\perp = -0.3$ implying that if the mean scale in the normal-adjacent-to-tumor tissue is less than or equal to exp($-0.3 - 0.05$) = 0.70 times the mean scale in the healthy tissue, then multiple significant gene sets are false positives.

We then applied the LFC Sensitivity Test to the same data set. As expected, we found that the LFC Sensitivity Test has low yet non-zero power. Zero significant gene sets were identified

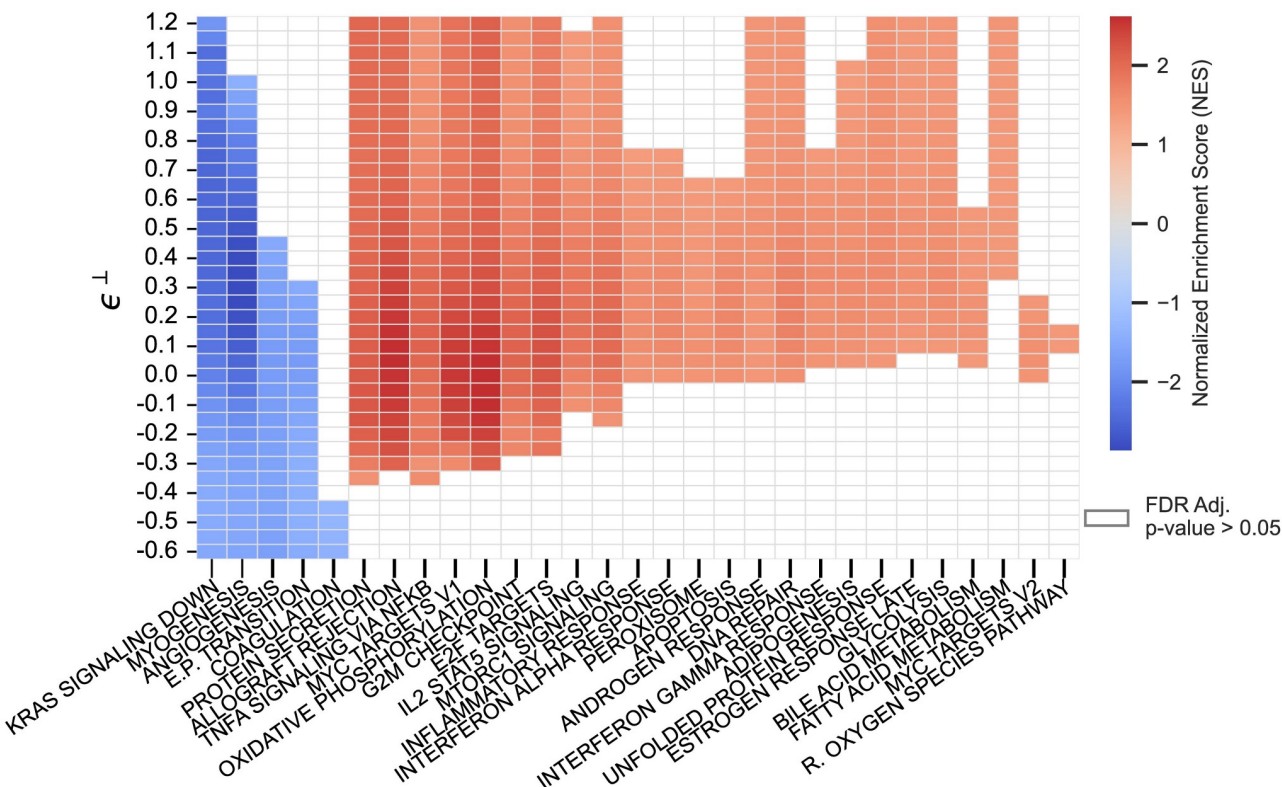

**Fig 2. In the context of real data, GSEA-LFC applied estimators can demonstrate substantial sensitivity to errors in scale assumptions.** LFC Sensitivity Analysis was used to replicate the results of Aran et al. [4], which used GSEA-LFC to compare differential pathway enrichment between normal-adjacent-to-tumor and healthy thyroid tissue. We explored sensitivity to errors in the CLR assumption which, for this study, equates to $\hat{\theta}^{\perp} = -0.3$. For an error $\epsilon^{\perp}$ (Y-axis), the implied true log-fold-change of scales is given by $\theta^{\perp} = \hat{\theta}^{\perp} + \epsilon^{\perp}$. Higher (or lower) values of $\epsilon^{\perp}$ correspond to a higher (or lower) scale in normal-adjacent-to-tumor compared to healthy tissue than assumed. The range of $\epsilon^{\perp} \in [-0.6, 1.2]$ was informed by prior research on how much scales can vary between conditions in similar experiments; it is asymmetric to account for the CLR assumption $\hat{\theta}^{\perp} = -0.3$ (see Methods for full details). Higher values (red) of the NES correspond to more enrichment in normal-adjacent-to-tumor tissue, and lower values (blue) more enrichment in healthy tissue.

when the LFC Sensitivity Test was applied to the 50 hallmark gene sets studied in Aran et al. [4]. However, expanding to a larger group of 4441 gene sets (see Methods) identified 96 gene sets as significantly enriched. In Section E in S1 Text we present a full power analysis which suggests that the power of the LFC Sensitivity Test ranges from 6% to 13.5% in the context of this data.

## GSEA with compositional weighting

Under the SRI framework, the distinction between whether a problem is scale reliant or scale invariant depends on the scientific goal (the target estimand). Until now, we have only considered target estimands defined by replacing estimated LFCs with true LFCs. For example, we defined the GSEA-LFC target estimand as $\phi_S = u(\theta)$ by replacing the estimates $\hat{\theta}$ with their true values $\theta$ in the GSEA-LFC applied estimator $\hat{\phi}_S = u(\hat{\theta})$. This approach allowed us to study how errors in scale assumptions, used in the estimation of $\theta$, propagate into the estimation of $\phi_S$. In this section, we take a different approach and assume that the applied estimator a researcher uses is tautologically consistent with their research goals. For a researcher using the

GSEA-LFC applied estimator, we assume that the target estimand is $\hat{\phi}_S = u(\hat{\theta})$. In this case, there are no scale assumptions, as there is no discrepancy between the methods applied and the goals of an analysis. Moreover, without scale assumptions, there is no need for sensitivity analysis. Instead, this section studies the research goals implied by this target estimand. This leads us to a deeper understanding of when researchers should be concerned about potential errors in scale assumptions.

To better express our meaning under this new approach, we modify the notation used in the prior sections. Rather than using $\hat{\theta}$ to denote estimated LFCs, where the superscript $\hat{}$ emphasized that this was an estimate, we now use the notation $\vartheta = \hat{\theta}$. The lack of a $\hat{}$ emphasizes that this is not simply an estimate or approximation but a quantity of direct scientific interest that is tautologically free of potential error. For the same reason, we replace $\hat{\phi}_S$ with $\varphi_S$. For a researcher using an applied estimator of the form $\hat{\phi}_S = u(\hat{\theta})$, we now have a target estimand of the form $\varphi_S = u(\hat{\theta}) = u(\vartheta)$. This notation also emphasizes that this target estimand is not the same as the GSEA-LFC target estimand in Eq (2). For concreteness, we again focus on the common use of the CLR assumption in LFC estimation: let $\vartheta$ denote the log-fold change of CLR transformed abundances (Eq 4). We call $\varphi_S$ the GSEA-CLR target estimand. How do we interpret the scientific goal represented by this estimand?

In Section F in S1 Text, we demonstrate that the LFC of CLR transformed abundances ($\vartheta_d$) is related to the LFC of the $d$-th entity ($\theta_d$) by the equation:

$$\vartheta_d = \theta_d - \frac{1}{D}\sum_{d=1}^{D}\theta_d.$$

In words, researchers purposefully choosing to analyze LFCs of CLR transformed abundances ($\vartheta_d$) (as opposed to LFCs of actual abundances, $\theta_d$) do not care if an entity is truly increased or decreased between conditions, only that the entity is increased or decreased relative to the average change of all of the entities. This distinction is shown visually in Fig 3. Extending this intuition to GSEA applied to $\vartheta$, researchers purposefully using this approach are only concerned about whether the entities of a set $S$ are non-randomly increased or decreased between conditions *relative to the average change of all the entities*. For example, such researchers would still be interested in an entity set that is not actually enriched (or depleted) between

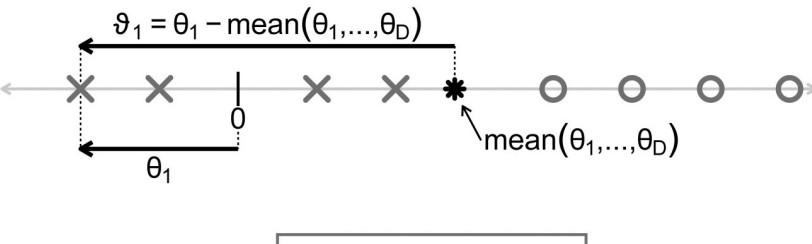

Fig 3. A visual depiction of the difference between Log Fold Changes (LFCs), $\theta$ defined by Eq (2), and LFCs of CLR transformed amounts, $\vartheta$ defined by Eq (4). The X's and O's are plotted on a number line and represent the LFCs of eight entities in and not in some set of interest. The index 1 refers to the leftmost X in the plot. $\text{mean}(\theta_1, \ldots, \theta_D)$ is the mean distance of all X's and O's from 0. In this illustration, none of the entities in the set (the X's) are strongly increased or decreased in amount between conditions: for $d \in S$, $\theta_d \approx 0$. Still, each of these entities has a negative $\vartheta_d$, as their LFCs are each less than the mean LFC of all the entities. According to the GSEA-CLR target estimand, this set would therefore be depleted whereas it is neither depleted nor enriched according to the GSEA-LFC target estimand. The equality $\vartheta_1 = \theta_1 - \text{mean}(\theta_1, \ldots, \theta_D)$ is derived in Section F in S1 Text.

conditions, but is only relatively enriched (or relatively depleted) when compared to the entities not in the set. Two examples will solidify this intuition and illustrate the practical implications of understanding this distinction.

Consider two researchers, Researcher A and Researcher B. Researcher A wants to identify if a particular genomic trait confers a selective advantage in a mixed microbial community exposed to an antibiotic. Researcher A does not care if the trait actually stimulates growth in the presence of the antibiotic, only that microbes with that trait increase in abundance *relative to microbes that do not have that trait*. This goal is more consistent with the GSEA-CLR target estimand than the GSEA-LFC target estimand. As a result, Researcher A can apply GSEA to LFCs estimated using the CLR assumption without needing to consider potential error in scale assumptions. In contrast to Researcher A, Researcher B wants to identify gene pathways that are differentially activated in diseased compared to healthy tissue. Researcher B wants to understand which pathways may play a causal role in the disease. Researcher B is not interested in a pathway that is unrelated to disease and is simply enriched relative to some other pathway that is repressed in disease. Researcher B's scientific goal is more consistent with the GSEA-LFC target estimand than the GSEA-CLR target estimand. As this is a scale reliant target estimand, we would recommend that Researcher B considers potential error in scale assumptions regardless of the applied estimator chosen.

Extending beyond the GSEA-CLR target estimand, Section F in S1 Text shows that the GSEA-CLR target estimand is actually just one of an infinite number of target estimands for DSA that are built around GSEA yet are scale invariant. We call this broader class *GSEA with Compositional Weighting* (GSEA-CW). Appealingly, each target estimand in this class can be naturally paired with a different applied estimator, which can in turn be computed from observed data. In Section F in S1 Text, we provide further characterization of this class to help researchers understand both the types of scientific goals that are invariant to erroneous scale assumptions and how to identify appropriate analytical tools given such goals. In Section G in S1 Text, we illustrate some of the advantages of this class over an alternative scale invariant approach to DSA recently proposed by Nguyen et al. [31].

## Discussion

Differential Set Analysis (DSA) is a core analysis performed in modern biomedical research [32]; it is used to identify sets of entities that are differentially enriched or depleted between two experimental conditions. Although the majority of our presentation focused on the analysis of gene expression data using GSEA applied to estimated LFCs, we also showed that these same problems can be found when studying 16S rRNA microbiome data or when investigating other popular methods such as CAMERA (Section A in S1 Text). In all these cases, we arrived at the same three conclusions: 1. for common scientific goals, even slight errors in scale assumptions can lead to false positives in DSA; 2. sensitivity analyses enable researchers to identify those gene and microbe sets whose significance is highly sensitive to errors in scale assumptions; 3. the sensitivity of DSA to errors in scale assumptions depends on the goal of the study.

Our results suggest caution is warranted when performing DSA from sequence count data as we do find that results can be highly sensitive to even slight errors in scale assumptions. For example, in the reanalysis of Aran et al. [4] we found that gene sets in the Apoptosis and Interferon Alpha Response pathways, which were reported by those authors as enriched (in Thyroid normal-adjacent-to-tumor tissue), were no longer significant with scale assumption error as small as $\epsilon^\perp = -0.05$. To be clear, we are not arguing that those conclusions are wrong: we do not know the ground truth LFCs. Instead, we have only studied the sensitivity of those

conclusions to errors in modeling assumptions. Still, we find such sensitivity concerning and suggest that such conclusions should be viewed skeptically.

The concept of target estimands highlights how the scientific goals of a study dictate the impact of erroneous scale assumptions. In this article we have discussed a number of target estimands such as GSEA-LFC and GSEA-CW. We also discussed a CAMERA based estimand in Section A in S1 Text. Still, we expect that some researchers will feel that they perform DSA for reasons not represented in the estimands we have studied. For example, researchers may perform GSEA on LFCs but be interested in population level LFC estimates, where instead of a mean in Eq (2), there is an expectation with respect to some population-level model. We believe the study of DSA from the perspective of different target estimands represents a prime area for future research.

This article has focused on DSA performed using sequence count data. Yet there has been an increasing interest in combating scale limitations by combining sequence count data with secondary measurements designed to directly measure the system scale. For example, in the study of human microbiota, some researchers use qPCR or flow cytometry to measure the total 16S rRNA concentration or the total cellular concentration in a sample [17]. While we believe that the field's increasing interest in these types of measurements necessitates more careful study, we suspect these measurements may be more limited than often discussed. Putting aside issues of the accuracy and precision of such measurements [14], we expect that the scale measured by these technologies may not accord with a researcher's desired notion of amount. Many researchers use qPCR to measure the total concentration of 16S rRNA in fecal material. In the context of DSA, how often are researchers interested in identifying if the concentration of 16S rRNA from microbes in fecal material is enriched or depleted? Even if this was a concentration measurement from the colon (rather than stool), we expect there are at least some researchers interested in defining enriched or depleted sets with regards to alternative notions of amount (different definitions of scale). As a result, we expect that these technologies will not solve the problem of scale limitations for all researchers. Neither do we claim that our methods are so general as to solve the problem of scale limitations. Still, we note that our methods are not tied to a single notion of scale and may therefore have some advantages over these external measurement-based approaches.

## Materials and methods

### Preprocessing and differential analysis of thyroid tissue RNA-seq data

Following Aran et al. [4], we downloaded pre-processed read count data for healthy and normal-adjacent-to-tumor tissue from the Genotype-Tissue Expression project (GTEx) and The Cancer Genome Atlas (TCGA) via NCBI's Gene Expression Omnibus (GEO) under accession numbers GSE86354 and GSE62944, respectively [33, 34]. The downloaded read count data were further processed to use the same 16038 genes, 361 healthy thyroid samples, and 59 normal-adjacent-to-tumor thyroid samples used by Aran et al [4]. LFCs were estimated using the Songbird multinomial logistic-normal regression model (Version 1.0.3) [15] including an intercept term and a binary condition indicator (healthy versus normal-adjacent-to-tumor). The model was trained with default parameters and validated by ensuring cross validation error plateaued over training epochs.

### LFC sensitivity analysis and testing of thyroid data

Following Aran et al. [4], GSEA $p$-values were calculated using a list of 50 hallmark gene sets from the Molecular Signature Database (MSigDB, Version 7.4.0) [35] and FDR corrected as

described in Subramanian et al. [1]. *p*-values were calculated using 25,000 entity set label permutations.

LFC Sensitivity Analysis was performed over the range $\epsilon^\perp \in [-0.6, 1.2]$ in increments of 0.05. This range of $\epsilon^\perp$ was chosen based on prior literature studying tumor versus normal tissue which suggested that total RNA abundance between these conditions could vary by as much as 2.5 fold [36]. Combining this range (±2.5 fold) with this dataset's CLR estimate of $\hat{\theta}^\perp = -0.3$ implied a range of $\epsilon^\perp$ of $\epsilon^\perp \in [-\log 2.5 - \hat{\theta}^\perp, \log 2.5 - \hat{\theta}^\perp] = [-0.6, 1.2]$.

The LFC Sensitivity Test was performed first on the 50 hallmark gene sets described above, and then on the MSigDB C2 (version 7.4.0) curated list of gene sets [1, 37]. Based on the default parameters of the GSEA software package [1], only gene sets that containing between 15 and 500 genes were retained in our analysis resulting in a set of 4,441 candidate gene sets. *p*-values for the LFC Sensitivity Test were calculated over range of $\epsilon^\perp \in [-200, 200]$ with a grid size of 1, except in the range $\epsilon^\perp \in [-10, 10]$ where a grid size of 0.1 was used for better resolution. In all cases a significance threshold of $p < 0.05$ was used.

## Simulated LFC sensitivity analysis using the positive predictive value

The simulated LFC Sensitivity Analysis presented in this work was summarized using the Positive Predictive Value (PPV). For each value of $\epsilon^\perp \in [-2, 2]$ considered, PPV was calculated as the percentage of entity sets for which $\hat{\phi}_s = \phi_s(\epsilon^\perp)$ when $\hat{\phi}_s \neq 0$ where $\hat{\phi}_s$ is the DSA estimate under the CLR assumption (when $\epsilon^\perp = 0$) and $\phi_S(\epsilon^\perp)$ denotes the true value of $\phi_S$ when the error in the LFC estimates is equal to $\epsilon^\perp$. It follows that, by definition, the PPV is equal to 100% when $\epsilon^\perp = 0$.

## Supporting information

**S1 Text. A document containing all supplementary sections mentioned in this work.** (PDF)

## Acknowledgments

We thank Rachel Silverman and Yen Duong for their manuscript comments.

## Author Contributions

**Conceptualization:** Kyle C. McGovern, Michelle Pistner Nixon, Justin D. Silverman.

**Data curation:** Kyle C. McGovern.

**Formal analysis:** Kyle C. McGovern.

**Funding acquisition:** Kyle C. McGovern, Justin D. Silverman.

**Investigation:** Kyle C. McGovern, Justin D. Silverman.

**Methodology:** Kyle C. McGovern, Michelle Pistner Nixon, Justin D. Silverman.

**Software:** Kyle C. McGovern.

**Visualization:** Kyle C. McGovern.

**Writing – original draft:** Kyle C. McGovern, Michelle Pistner Nixon, Justin D. Silverman.

**Writing – review & editing:** Kyle C. McGovern, Michelle Pistner Nixon, Justin D. Silverman.

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
