## [Decision Letter · Decision Letter 0]

21 Aug 2023

Dear Dr. Silverman,

Thank you very much for submitting your manuscript "Addressing Erroneous Scale Assumptions in Microbe and Gene Set Enrichment Analysis" for consideration at PLOS Computational Biology. As with all papers reviewed by the journal, your manuscript was reviewed by members of the editorial board and by several independent reviewers. The reviewers appreciated the attention to an important topic. Based on the reviews, we are likely to accept this manuscript for publication, providing that you modify the manuscript according to the review recommendations.

Overall the reviewers were positive about the manuscript. The reviews do indicate some corrections that need to be made, most of which are minor and in regard to the clarity and completeness of the reporting.

Sincerely,

Nic Vega, Ph.D.

Academic Editor

PLOS Computational Biology

Jian Ma

Section Editor

PLOS Computational Biology

Overall the reviewers were positive about the manuscript. The reviews do indicate some corrections that need to be made, most of which are minor and in regard to the clarity and completeness of the reporting.

Reviewer's Responses to Questions

**Comments to the Authors:**

Reviewer #1: As the authors stated in the manuscript, they are proposing a sensitivity measurement and a statistical test for controlling false positive rates of differential set analysis, potentially due to technical factors such as limitations of the sequence counting measurements. However, the contents of the manuscript should be reorganized to clearly distinguish their own contributions and the novelty from the previous results. This makes the paper difficult to read although the problem considered in this work mathematically sounds and could be attractive to biomedical research. I think the approach could be described better, first at a high-level and then also formally. In the application to real data, what is conclusion related to biological knowledge?

Reviewer #2: This is a well written and reproducible article addressing an important practical problem from a rather theoretical perspective while providing practical solutions.

As a general comment, I would say that the mathematical notations are quite hard to follow (even for someone with a mathematical background). It may be useful to include a table with an overview of these notations with corresponding definitions.

Regarding the structure of the manuscript, I understand that the proposed methods are actually the results of this methodological article. However, I find the structure of the paper a bit confusing, since the Results section contains both the description of methods and numerical results. A structure with the (new) methods before the results would appear more appropriate - if allowed by the journal's policy.

Specific comments:

- Please define the term "compositional survey" the first time it occurs.

- "a unique estimate of \\hat{\\Psi}": the authors probably mean \\Psi rather than \\hat{\\Psi}?

- It is not clear to me why Eq.(5) results from a simplification of the assumption. Please explain.

- On p6 it is stated "GSEA-LFC uses GSEA to test whether the LFCs of entities in the set cluster together non-randomly when compared to the LFCs of entities not in the set", suggesting that the authors consider competitive hyotheses. They could mention that this deviates from standard GSEA, that considers self-contained hypotheses.

- p7: "We assume the target estimand and estimator are given by" -> target estimand and estimator cannot be given by the same expression. Also, I do not understand why we need tilde to denote the true value. This does not fit usual conventions in statistics.

- p7: I do not understand the rationale of the assumption \\epsilon \\approx \\epsilon^\\perp.

- Please check the references carefully. Some of them are incomplete/incorrect (esp. article numbers of online journals).

Reviewer #3: The authors build on previous work to show that errors in scale assumptions can lead to high rates of type-I errors in differential set analysis, and study the impact of scale assumptions on a well-known approach to gene set enrichment analysis.

The authors develop a sensitivity analysis to help determine type-I errors and give examples of research goals that are not sensitive to errors in scale. They demonstrate how errors in scale assumptions can affect analyses - on simulated data, they find that the false positive rate can be above 50% with modest errors. On real data they find that with plausible error rates the enrichment of some gene sets previously identified as enriched may not be statistically significant.

The paper provides a valuable guide to researchers to raise awareness of and determine when their analyses could be sensitive to errors in scale assumptions, and serves as a basis for further research in this area. It is thorough and well written. I just have a few minor comments:

Main Points:

• In the abstract the headline figure for false positive rates is 70%, but I’m not sure what part of the text this is referring to? Is it the analysis of simulated data?

• The “CLR assumption” is used in the analysis of real data and simulated data to demonstrate the main point of the paper, so I think it would be helpful to cite other work where this assumption has been used.

• The derivation of equation (5) is not clear to me. It would be helpful if this could be added (possible to the supplementary materials), and a description of the proportional and scale components of v added to the text.

• It would be helpful to have a description of what the plausible error is in terms of the experiment parameters for section 2.4. There is some discussion of this in section 4.2 but I think it would be helpful to give an example in 2.4, i.e. what does a scale error of -0.1 translate as in terms of the experimental data?

• The discussion says that “sensitivity analyses provide a powerful tool for mitigating these problems”. I’m not sure that this has been demonstrated. Similarly, the next sentence says “this work provides tools to mitigate the impact of erroneous scale assumptions”. Certainly it provides tools to identify them and therefore avoid them, or at least acknowledge them, but I’m not sure what the tools provided that mitigate them are. Perhaps this could be reworded to be a little clearer.

Minor Points:

• There are several typos throughout the text, including places where math-mode is not used but I think it should be (e.g. for “p-value” and “n=92” in the caption for Figure S1). I leave a list of these for the copy editor.

• There are several cases where the same paper is cited many times, even in the same paragraph. I think it is sufficient to give the citation only the first time that the paper is mentioned.

• The abstract states “Second, we introduce a statistical test that provably controls Type-I error at a nominal rate”, yet in the text the exposition of the statistical test is relegated to the supplementary materials. If this part of the abstract is supposed to mirror the outline of the paper then maybe it should say something like “Second, we demonstrate on real and simulated data the Type-I error that can occur with incorrect scale assumptions.”?

• In the first paragraph of Section 1.1.1 I think “DNA molecules” should be replaced with “DNA reads” or “DNA sequences”.

• In the first paragraph of Section 1.1.2, “…that produces a unique estimate of $\\hat{\\psi}$ must have..”, I think the hat should be removed.

• I don’t think the definition of $W_{dn}^{clr}$ after equation (4) is consistent with its use in supplementary section 6.1. It seems like there are too many logs (i.e., one in equation (4) itself and one in the definition of $W_{dn}^{clr}$). Is this right?

• In section 1.1.2, the abbreviation CLR is used before it is defined.

• In the first paragraph of Section 2.3 ,“The GSEA algorithm is visualised in Figure 1…” – I don’t think it is. Is this referencing a figure that is no longer present?

• I found the arrows and text in Figure 3 a little bit confusing. I understand what they are trying to demonstrate but I’m not sure that it is correct.

• In Supplementary section 1, second equation, should $\\alpha_n^\\perp$ be $1$ when $x_n = 0$? In the text following it looks like it should be 0.

• In Supplementary section 4, the text says, “The facultative anaerobic (F. Anaerobic) microbe set, on the other hand, was highly sensitive as it was only enriched at a single value.” I’m not sure that I agree with describing this as highly sensitive – surely it is the opposite?

• In Supplementary section 6.1 a detailed derivation is given (which is great). But the first step where $W_n^\\perp$ cancels is not shown explicitly.

• I’m not sure what $n = [1,2]$ means in the second paragraph of supplementary section 7.1.

**Have the authors made all data and (if applicable) computational code underlying the findings in their manuscript fully available?**

Reviewer #1: Yes

Reviewer #2: Yes

Reviewer #3: Yes

PLOS authors have the option to publish the peer review history of their article (what does this mean?). If published, this will include your full peer review and any attached files.

Reviewer #1: No

Reviewer #2: No

Reviewer #3: No

Figure Files:

Data Requirements:

Reproducibility:

References:

---

## [Decision Letter · Decision Letter 1]

4 Nov 2023

Dear Dr. Silverman,

We are pleased to inform you that your manuscript 'Addressing Erroneous Scale Assumptions in Microbe and Gene Set Enrichment Analysis' has been provisionally accepted for publication in PLOS Computational Biology.

Best regards,

Nic Vega, Ph.D.

Academic Editor

PLOS Computational Biology

Jian Ma

Section Editor

PLOS Computational Biology

Reviewer's Responses to Questions

**Comments to the Authors:**

Reviewer #2: The authors adequately addressed my concerns.

Reviewer #3: The authors have made some significant changes to the layout of the paper and addressed all of my comments from the first review. The paper is well written and the clarity of the exposition has been markedly improved. I think this is an interesting paper and I recommend it for publication. I only have one more tiny comment which is that "DNA sequences" on line 22 should be "DNA molecules" (as in the original text).

**Have the authors made all data and (if applicable) computational code underlying the findings in their manuscript fully available?**

Reviewer #2: None

Reviewer #3: Yes

PLOS authors have the option to publish the peer review history of their article (what does this mean?). If published, this will include your full peer review and any attached files.

Reviewer #2: No

Reviewer #3: No

---

## [Editor Report · Acceptance letter]

15 Nov 2023

PCOMPBIOL-D-23-00819R1 

Addressing Erroneous Scale Assumptions in Microbe and Gene Set Enrichment Analysis

Dear Dr Silverman,

I am pleased to inform you that your manuscript has been formally accepted for publication in PLOS Computational Biology. Your manuscript is now with our production department and you will be notified of the publication date in due course.

With kind regards,

Zsofi Zombor
